# Targeting Smartphone Use While Driving: Drivers' Reactions to Different Types of Safety Messages

**Laura Šeibokaitė \*, Rasa Markšaitytė, Auksė Endriulaitienė** **, Justina Slavinskienė, Dainora Šakinytė and Gerda Tamajevaitė**

Psychology Department, Vytautas Magnus University, 44248 Kaunas, Lithuania; rasa.marksaityte@vdu.lt (R.M.); aukse.endriulaitiene@vdu.lt (A.E.); justina.slavinskiene@vdu.lt (J.S.); dainora.sakinyte@stud.vdu.lt (D.Š.); gerda.tamajevaite@stud.vdu.lt (G.T.)
* Correspondence: laura.seibokaite@vdu.lt; Tel.: +37-037327824

**Abstract:** Only a few previous studies analyzed the effectiveness of road safety messages targeting smartphone use while driving and only several of them used messages from an ongoing road safety campaign. Thus, contributing to the field, this study aimed at testing the effectiveness of two types of social messages (threat appeal and threat appeal together with safe behavior role modelling) targeting smartphone use while driving. Ninety-three drivers were randomly assigned to two experimental ($n1 = 26$; $n2 = 37$) and one control ($n = 29$) groups. Each experimental group was presented with one 30 s length video message to reduce or stop smartphone use while driving. Messages differed in terms of threat appeal and modelling of safe behavior. The control group was presented with a 30 s length video clip showing neutral driving related content. The results revealed that threat appeals (alone or together with a safe role model) resulted in less positive emotions when compared to the control group's reported emotional reactions. The message with threat appeal only also resulted in more negative emotions compared to the control group. With regards to behavioral intentions, road safety messages used in this study had minor effectiveness: the threat appeal message reduced the intentions to use smartphones while driving, only when previous behavior has been controlled. In sum, messages targeting smartphone use while driving were effective at least to some extent in changing drivers' emotions and intentions not to be involved in targeted behavior, but the effect was minor and threat appeal only showed higher effectiveness.

**Keywords:** smartphone use while driving; road safety message; threat appeal; emotional reactions; behavioral intentions

## 1. Introduction

Many studies confirmed that illegal smartphone use while driving impairs driving performance [1–3] and contributes significantly to the risk of traffic accident or injury [4–6]. However, the use of smartphones became an integral part of everyday life due to the possibility of instant communication with others and easy access to information searches or social networking [7–9]. Consequently, the number of drivers that text, talk or access the web while driving is still growing [5,10,11]. Thus, distracted driving due to in-vehicle smartphone use has become an increasing issue of road safety [12].

### 1.1. Effectiveness of Safety Messages Targeting Smartphone Use While Driving

Enhanced public awareness through social marketing or educational campaigns might be effective means of the prevention of risky smartphone use while driving [13]. Unfortunately, a large part of the knowledge on the effectiveness of road safety campaigns was based on the research using messages targeting speeding and drunk driving. The research on the effectiveness of road safety messages targeting smartphone use while driving is scarce. Hayashi and colleagues found that threat appeal significantly reduced drivers'

intentions to text while driving and improved their attitudes towards texting while driving [14]. Gauld and colleagues developed six road safety messages targeting different modes of smartphone use while driving [11,15]. In order to increase persuasiveness and address specific beliefs of a driver, all messages were developed in accordance with the Step Approach to Message Design and Testing. However, the results revealed that only messages targeting monitoring or reading information on a smartphone decreased the intention to engage in this behavior in the future [15] and the effect for young female drivers was higher compared to male drivers [11]. Both positive and negative messages were found to be effective. Still, it should be noted, that Gauld and colleagues used written descriptions of possible video messages in both studies [11,15].

Thus, there is still a lack of knowledge about the possible effectiveness of safety messages targeting smartphone use while driving that are already used in road safety campaigns and are based on general principles of social marketing. Therefore, the aim of this study was to test the effectiveness of two types of messages targeting smartphone use while driving among Lithuanian drivers from an ongoing road safety campaign: threat appeal and threat appeal with safe behavior role modelling. The effectiveness of road safety messages was assessed by two outcome measures: self-reported emotional reactions and behavioral intentions to reduce the targeted behavior.

### 1.2. Emotional Arousal as Key Component of Effective Road Safety Video Messages

As research evaluating the effectiveness of video messages targeting smartphone use while driving is rare, it might be important to review the lessons that have been learned for the studies dealing with other risky behaviors on the road. Almost all studies were conducted using experimental design; therefore, they allow drawing conclusions about causal effects. Fear-arousing threat appeals are the most commonly used in road safety campaigns [16–19]. Still, the research on threat appeals in road safety messages has provided controversial evidence of their effectiveness. Several previous studies confirmed that a stronger level of threat predicted higher effectiveness of the road safety message [20–22]: one of them confirmed strong emotional arousal which led to decreased speed while driving in the simulator [20,22]; another drew the conclusion that appeals evoked fear but did not impact driving outcomes from the meta-analysis of studies conducted during the 1990–2011 period [21]. Rhodes showed that only medium-strong fear reduced self-reported intentions to speed [23]. Other authors reported that threat appeals evoked using a different modality of messages (videos, leaflets or combination of two) had no effect on intentions to drive slower based on the self-reports of participants [18]. In other studies, positive appeals were found to have a greater effect on intentions for safer driving compared to threat appeals in road safety messages [17,19,24]. In Lithuania, positive appeals are hardly used for promoting safe driving, thus, this methodology of persuasion is out of the scope of the current paper.

According to some authors, threat appeals showed greater short-term persuasiveness while positive appeals were found to be more effective over time [25,26]. Such contradictions in research results may be explained by the fact that emotions influence real driving behavior indirectly. Many authors agree that the underlying mechanism in the relation between emotion and behavior is that emotional appeals arouse tension that motivates further processing of the message and leads to changes in attitudinal and self-efficacy beliefs [16,19,20,23,27,28]. Consequently, according to the Theory of Planned Behavior, attitudes and beliefs should increase intentions to behave more safely and intentions could turn into real driving behavior [29]. Not always do intentions result in real driving behavior as many other variables may be important in a particular situation [26,28,30,31], still, driving intentions were found to be the strongest predictor of driving behavior in many previous studies [32,33]. As road safety messages with either negative or positive emotional appeals were found to be more effective compared to rational argumentations [16,18–20,24], emotional appeals were acknowledged as the key element of the possible effectiveness of safety messages in this emotion–cognition–intention–behavior relationship [19].

*1.3. The Importance of Positive Behavior Modelling in Safety Messages*

Additionally, threat appeals could evoke a range of negative or positive emotions which might reduce the persuasiveness of the message [34,35]. Algie and Rossiter indicated that the feeling of threat may change during the video clip [16]. The recommendation of safer driving behavior at the end of the road safety message decreased the perceived fear [16] and decreased the need to cope with strong negative emotions [36,37]. Besides, the role modelling of safer behaviors helping to avoid negative consequences of risky driving was found to be a more important predictor of the effectiveness of the road safety message compared to threat appeal only [16,17,38]. Thus, it was hypothesized that the safety message presenting just the potential threat of smartphone use while driving aroused stronger negative and fear emotions and weaker positive emotions compared to the threat appeal with demonstration of safer driving behavior. However, threat appeal with safe behavior modelling was expected to result in a greater decrease in intention to use a smartphone while driving compared to threat appeal only.

*1.4. Novelty and Relevance of Current Study*

To sum up, some aspects that highlight the novelty, relevance and contribution of this study should be mentioned:

- the study focuses on the effectiveness of safety messages targeting smartphone use while driving, whereas other studies mostly use messages targeting drunk driving and speeding;
- two safety messages—threat appeal only and threat appeal together with safe behavior role modelling—are compared in this study;
- the importance of previous illegal smartphone use while driving behavior is controlled when evaluating the effectiveness of the safety message.

## 2. Materials and Methods

*2.1. Participants*

This study enrolled three randomly assigned groups of participants: two experimental and one control. The total sample of 92 drivers participated in the study (Experimental 1 *n* = 26; Experimental 2 *n* = 37; Control = 29). All participants were students (22.8 percent males) and possessed a B category license allowing driving motor vehicles with a mass not exceeding 3500 kg. Age range was between 18 and 50 years with the mean age 25.16 years (SD = 7.0). The driving experience ranged from 1 to 26 years (M = 5.9, SD = 6.0). A total of 48.9 percent of participants drove daily, 13 percent drove less than once a week. None reported being caught and fined for illegal smartphone use while driving.

*2.2. Stimuli*

The experimental design was employed for the study, where the independent variable was the video message to reduce or stop smartphone use while driving. Messages differed in terms of threat appeal and modelling of safe behavior (the study design is presented in Figure 1). Three different 30 s video messages were used. Two road safety messages created by the Lithuanian Road Administration under the Ministry of Transport and Communications with permission were used in the study. Both were demonstrated on TV or the internet before the study. Safety messages were selected according to several criteria: hand-held smartphone use while driving must be a target of video clip; young people must be actors in the scenes; threat appeal must be central in a message; actors must not talk; the clip length must be 30 s.

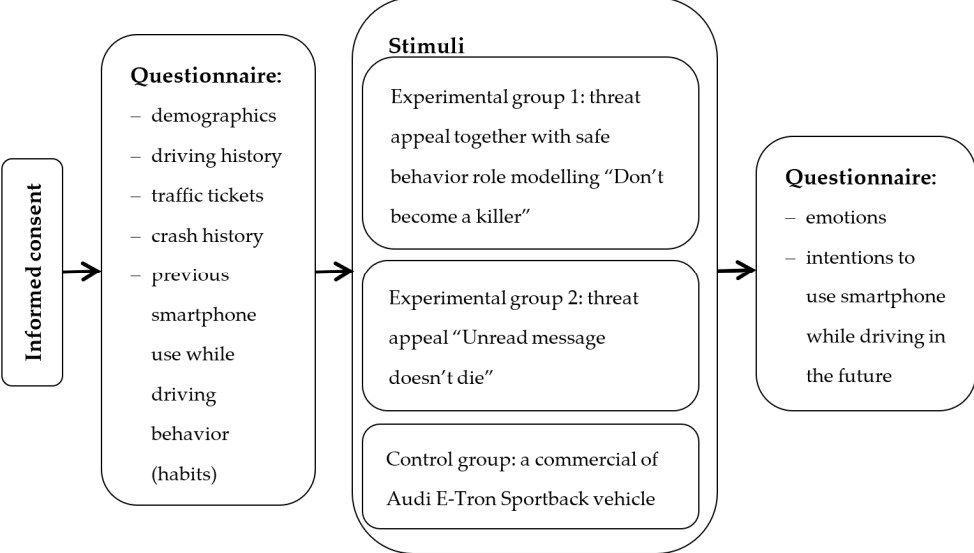

**Figure 1.** Study design.

Experimental group 1 was exposed to the clip called "Don't become a killer". The story involves a young woman who is driving with small kids in the back seat. She picks up her phone when it starts to ring. When the camera moves toward her face, the person with a killer's mask is talking on the phone instead of her. A voice behind the scenes says: "Don't become a killer. One out of four accidents happens due to smartphone use while driving". The killer hangs up the phone, and the smiling woman appears instead and proceeds with driving. We treated this message as possessing a threat appeal as well as having a model of proper behavior instead of responding to the phone call. The safety message could be watched on youtube.com at https://rb.gy/hvwby5 (accessed 22 October 2021; in Lithuanian).

The participants in experimental group 2 watched the clip called "Unread message doesn't die". The clip shows two young drivers: male and female operating different vehicles and texting while driving. Both drivers get messages and respond to them by texting. They approach the same intersection where a little girl is going through the crossing. The drivers do not look at the road. The observers might hear sliding tires when breaks are hit sharply and crashing vehicles. Finally, only a shoe of the girl is visible in front of the stopped vehicles. A voice behind the scenes says: "Unread message doesn't die. Choose life". The message contains a threat, but it does not demonstrate desirable behavior. The safety message could be watched on youtube.com at https://rb.gy/vt21ic (accessed 22 October 2021; in Lithuanian).

It needs to be mentioned that both safety messages were focused on different modes of smartphone use while driving: answering a phone call and texting. Still, many authors reported that texting and talking on the phone are the most common behaviors while driving [5,10,39,40], with both having a significant negative effect on driving performance [2]. All modes of smartphone use while driving are understood as common and socially acceptable behavior for drivers too [5,10,37,41,42]. Thus, the authors believe that the content of the message (threat appeal only and threat appeal with safe behavior modelling) was a more important factor than the mode of smartphone use behavior. This is especially the case because previous research confirmed that different content of safety messages had pretty much the same effects for different risky driving behaviors [35] or even other risky behaviors [31,38]. On the other hand, the authors had a very limited choice of safety messages from ongoing traffic safety campaigns in Lithuania.

It was decided to use the scene of driving a vehicle for the control group as well. Participants of this group watched a commercial about the Audi E-Tron Sportback vehicle where a young man drives a car in an urban area with a male child in the back seat. The

child is peacefully playing with a puzzle toy. The driver is relaxed, enjoying the driving process. The clip demonstrates some smart features of the vehicle like automated parking, spotting of traffic lights and pedestrians. Cheerful music is playing throughout the clip. No threat is anticipated. The clip is not directly safe behavior-related either.

*2.3. Measures*

2.3.1. Emotions

Current emotions were assessed by the list of emotions used in previous research [15]. It included emotional reactions like sad, fearful, anxious, annoyed, relaxed, competent, happy, proud, excited, amused, flattered, agitated and relieved. Items "angry" and "disgusted" were added to the list as well as the authors assumed that both could be related to the content of commercials. Participants had to evaluate how strongly they currently felt each of these on a five-point scale.

The list of emotions was factorized, and a Varimax rotation was applied. Observed data were suitable for factorial analysis (KMO = 0.836). A three-factor solution was accepted as the most appropriate both statistically and theoretically. Three factors explained 65.27 percent of variance. The first factor was named "positive emotions" and was composed of 7 items (excited, amused, proud, flattered, happy, competent and relaxed; Cronbach alpha—0.87). The second factor included five items: angry, annoyed, agitated, disgusted and sad. It was titled as "negative emotions" (Cronbach alpha—0.85). The third factor included: fearful, anxious and relieved. However, item "relieved" was removed as it had similar loadings across two factors and made the factor unexplainable. So, for further use only two items composed the last factor with the title "fear/anxiety" (Cronbach alpha—0.67). It was decided to use this factor for further analysis due to the rationale of road safety messages to induce fear about one's safety and as a result change the behavior. Values of the scales represent the averaged answers to items that were included in the scale. The higher score of the scale indicated the more expressed emotional state.

2.3.2. Intentions

Respondents were asked to rate the possibility to use smartphones while driving for the different purposes in a five-point scale (1—not likely, 5—most likely). Items included reading SMS and messages in social media; texting; writing messages and commenting on social media; speaking into the hand-held phone; scrolling on social media and searching for information on the internet. The four-item scale of intentions to use smartphones while driving was composed (Cronbach alpha—0.85). A higher score indicated higher intentions.

Emotional state and intentions to use a smartphone while driving were used as dependent variables and were the measures of safety message effectiveness.

2.3.3. Smartphone Use While Driving Habits

Smartphone use while driving in the past was measured by five items indicating reading SMS and messages on social media; writing SMS, messages and commenting on social media; calling others by hand-held phone; responding by hand-held phone; scrolling on social media and searching for information on the internet. Participants were asked how frequently they performed each behavior. The scale had good internal consistency (Cronbach alpha—0.80). A higher score implied more frequent smartphone use while driving in the past.

The questionnaire also included questions regarding the participant's gender, age, driving experience and frequency, number of kilometers driven per week, participation in car crashes and being punished for any traffic rule violations and smartphone use in particular.

*2.4. Procedure*

Participants were recruited by advertising the study via the university's webpage and emails that were sent to all students in one university. Research was conducted in the

laboratory where conditions for each participant were created as identical as possible. In the beginning of the research session, participants were randomly assigned to one of the experimental or control groups and were informed about the nature of the experiment and procedure. They read and signed an informed consent form. Then, participants answered several demographical questions and indicated their driving history, previous smartphone use while driving and participation in crashes. Then, participants watched the video clip and again filled in the questionnaire, where emotional reactions and intentions to use the smartphone while driving were measured. The study design is presented in Figure 1.

The success of randomization was assessed by evaluating the differences in several variables among the experimental and control groups. Results revealed that groups did not differ significantly according to participants' gender, age, driving experience and exposure and habits of mobile phone use while driving ($p > 0.05$). The only statistical tendency observed was towards the control group riding more kilometers per week than experimental groups ($\chi^2 = 14.641$; df = 8; $p = 0.07$). Based on the results it might be concluded that groups were composed on a random basis and assumed as equal before intervention.

Students participated in the study on a voluntary basis, no incentives were proposed. The study plan was approved by the Institutional Review Board (permission No. EKL-2019.01).

### 2.5. Statistical Analysis

The normality of distribution tests was run for all dependent variables. The scale of positive emotions was distributed according to normal distribution (Kolmogorov–Smirnov—0.086; $p = 0.083$). Other scales violated normal distribution significantly ($p < 0.001$). Values of skewness and kurtosis ranged between −1 and 1 for the fearful/anxious scale; therefore, the distribution was treated as close to normal and parametric statistics were applied. Skewness and kurtosis exceeded 1 (ranged between 1.07 and 1.44) for scales of intentions to use smartphones while driving and negative emotions. After logarithmic transformation values fitted to range of −1 and 1, thus the normality assumption was accepted, parametric tests were applied across all data sets.

A one-way ANOVA was applied to evaluate the effectiveness of a specific safety message on positive and negative emotions as well as intentions to use smartphones. For the in-depth analysis of the effect of previous smartphone use behavior for intentions in all groups the univariate analysis of variance was run.

### 3. Results

### 3.1. The Impact of a Safety Message on Emotions

In order to evaluate the impact of a safety message on emotional state a one-way ANOVA was applied. Three scales of emotional states after watching the video clip were compared in two experimental and one control groups (Table 1). Results revealed that the control group reported more positive emotions after watching the clip than any of the experimental groups. In terms of negative emotions, the significant difference was observed only between experimental 2 and the control group: students in the experimental 2 group expressed more negative emotions. There was a statistical tendency that the control group reported less fear and anxiety after watching the clip compared to both experimental groups. None of the emotion-related scales differentiated in the experimental 1 and 2 groups.

**Table 1.** Comparison of emotional state and intentions to use a smartphone while driving among groups.

| | | Positive Emotions | Fear/Anxiety | Negative Emotions (Transformed) | Intentions to Use Smartphone (Transformed) |
|---|---|---|---|---|---|
| Experimental 1 | Mean | 2.12 | 2.12 | 0.38 | 1.88 |
| | Std. Deviation | 0.70 | 0.94 | 0.40 | 0.45 |
| Experimental 2 | Mean | 1.92 | 2.09 | 0.54 | 1.85 |
| | Std. Deviation | 0.66 | 0.89 | 0.41 | 0.41 |
| Control | Mean | 3.12 | 1.57 | 0.18 | 2.00 |
| | Std. Deviation | 0.76 | 0.74 | 0.31 | 0.48 |
| ANOVA statistics | F | 25.97 | 3.83 | 7.19 | 1.03 |
| | df | 2 | 2 | 2 | 2 |
| | $p$ | <0.001 | 0.025 | 0.001 | 0.360 |
| Pairwise comparisons [1] | E1/E2 | 0.539 | 0.996 | 0.332 | 0.964 |
| | E1/C | <0.001 | 0.069 | 0.141 | 0.601 |
| | E2/C | <0.001 | 0.053 | <0.001 | 0.382 |

[1] Dunnett T3 criterion was used for negative emotions due to unequal variances, for other comparisons Scheffe criterion was applied.

*3.2. The Impact of the Safety Message for Intentions to Use Smartphone While Driving*

The same statistics as in the case with emotions were used to evaluate the impact of the message on intentions to use smartphones while driving. The experimental and control groups did not differ in terms of intentions (Table 1).

As it is known from the large body of literature that behavioral or intentional changes are strongly related to previous behavior [11,15], it was decided to control smartphone use while driving habits in the analysis. The univariate analysis of variance was performed, where intentions were treated as the dependent variable; group and smartphone use habits' scale were input as predictive factors. Results revealed that the observed model explained 63.5 percent of intentions to use a smartphone. Previous behavior was the most significant predictor and accounted for the biggest part of variance (Table 2). There was only a statistical tendency indicating that group differences could explain intentions to use smartphones while driving in the future. The variable "group" added only 3.5 percent of variance to the explanation. A pairwise comparison of intentions among groups showed that participants of the control group had higher intentions to use smartphones while driving than their peers in experimental 2 group ($p$ = 0.045) when previous behavior was controlled. However, the difference between the control and experimental 1 groups was not observed ($p$ = 0.188). It might be concluded that after controlling for previous phone use while driving habits, the safety message with threat appeal only had a small short-term effect, if any, for intentions to reduce smartphone use while driving.

**Table 2.** Univariate analysis of variance for intentions to use smartphones while driving.

| Source | Type III Sum of Squares | df | Mean Square | F | Significance | Partial Eta Squared |
|---|---|---|---|---|---|---|
| Intercept | 4.998 | 1 | 4.998 | 69.672 | <0.001 | 0.442 |
| Smartphone use habits | 11.168 | 1 | 11.168 | 155670 | <0.001 | 0.639 |
| Group | 0.229 | 1 | 0.229 | 3.188 | 0.078 | 0.035 |

**4. Discussion**

Although the effectiveness of road safety campaigns is widely investigated in the field of speeding and drunk driving, limited data are available about how they work in reducing smartphone use while driving specifically. This study aimed to test the effectiveness of two types of safety messages targeting smartphone use while driving among Lithuanian drivers (in terms of two outcome measures: self-reported emotional reactions and behavioral intentions). It was hypothesized that a safety message presenting just the potential threat of smartphone use while driving would arouse stronger negative and fear emotions and

weaker positive emotions compared to the threat appeal with demonstration of safer driving behavior. However, threat appeal with safe behavior modelling was expected to result in a greater decrease of the intention to use a smartphone while driving.

The results partially supported the above hypotheses. In line with previous research [11,24], it was found that safety messages targeting smartphone use while driving were effective in evoking the emotional reactions of drivers. It was confirmed that threat appeals resulted in less positive emotions. The message with threat appeal only resulted in more negative emotions. There was also a tendency for both safety messages to cause more fearful emotions. Therefore, it might be expected that these emotions, if remembered, will provide some guidance and control in subsequent risky driving behavior or motivate behavior change [22]. Contrary to the results reported by Algie and Rossiter [16] and Rait [37], no difference was found in emotions based on the type of the road safety message. Both messages—only threat appeal and threat appeal together with positive role modelling—resulted in similar levels of observers' emotions when compared to the control group. This might be explained by methodological reasons in terms of assessing emotions with a self-report instrument. Some authors argue that self-evaluation of emotions might not be very precise and should be complemented by other types of measurement [20], as people may not be reflective of their emotions and provide biased data [43]. Another reason might be the severity of reported emotions. In the current study, drivers reported quite low scores of experienced emotions after watching video messages in both experimental groups. This in some sense might support the Terror Management Theory [23,44] which suggests that people cope with negative emotions using a variety of defense mechanisms if they are presented with explicit messages involving mortality. During the denial of fear, other emotional regulation processes, like re-appraisal of the threat after positive stimuli, might not work. In addition, according to the Extended Parallel Process Model (EPPM) [45] when people deny emotions and appraise a threat as low, they are not likely to process the message further and to respond to positive models. Finally, we did not control for drivers' familiarity with the advertisements and exposure to them before participation in the study. Previous research revealed that familiar hazards are usually perceived as less threatening; therefore, they evoke fewer negative emotions and do not encourage further focus or processing [46].

Regarding behavioral intentions it might be concluded that road safety messages used in this study had a minor effect. As seen in Plant et al. and Hayashi et al. it was revealed that threat appealing messages reduced the intentions to use smartphones while driving, only when previous behavior has been controlled [14,26]. However, contrary to Rait, the message providing role modelling of safe behavior was not effective and did not contribute to lower smartphone use intentions [37]. Nevertheless, as a discrepancy with previous research, the assumption that road safety messages with positive role models might be more effective and more appropriate in road safety campaigns rather than purely threat evoking appeals was not supported [16,17,38]. If we assume that threat appealing messages influence driving behavior through the underlying mechanism of emotions, some discrepancies from previous research might be explained by cross-cultural differences. Usually, it is acknowledged that emotional reactions are culture and country specific [47]; therefore, drivers from countries with different traffic safety climates may respond to road safety messages differently [48]. Clearly this is just the exploratory explanation and should be investigated in future research.

The present investigation has limitations that should be considered before applying the results to different contexts. Although the participants were enrolled in experimental groups randomly, non-random sampling of the whole study sample does not let one generalize the conclusions to larger driver populations. The experimental design of the study yielded some plausible conclusions about cause–effect relationships, still the small sample size and self-report measurements of intentions and emotions encourage cautious application of these data. In addition, very limited exposure of participants to road safety intervention (watching one video clip) decreases the possibility to explore the real

cumulative persuasiveness of road safety messages more extensively. It is hardly feasible that the driver will be exposed to the message only once in real life. The present study provides only limited answers about the effectiveness of road safety appeals, as only short-term effects were focused on. Finally, there is always discussion in the literature of whether the change in intentions or emotions is the indicator of real behavior change [26,28,30,31]. In this study, the outcome variables were distal and indirect measures of behavior change, still acknowledging that emotions have the impact on cognitions and subsequently on driving behavior [20]. In addition, the messages used in the study were of slightly different natures: one was focused on texting while another was on answering a hand-held phone. Gauld et al. proposed the idea that road safety messages are hardly comparable if they focus on qualitatively different behaviors indicating distracted driving [11,15].

Despite the mentioned limitations, there are several implications that emerge from this study. The findings confirm that safety messages targeting smartphone use while driving have at least some potential to change drivers' emotions and intentions not to be involved in this risky behavior. The findings replicate similar results from other fields of traffic safety campaigns, like speeding, drunk driving and seat belt use, supporting the idea that diverse risky driving behaviors share common underlying mechanisms and may be targeted effectively with social interventions. The importance of past habits to take risks on the road is so influential that it must not be ignored in future research and practical implementations when predicting future behavior of the driver. Still, the main contribution of this study is related to the result that threatening messages have a higher potential to change the smartphone use while driving behavior of Lithuanian drivers. It seems that stronger negative emotions aroused by a threat appeal draw the attention of the driver and make him or her to acknowledge some dangers of smartphone use while driving. As a result, the intention to behave in this risky manner decreases slightly. This encourages the further use of messages with threat appeal only for smartphone use in traffic safety campaigns, while the acceptance and effectiveness of other modes of social messages (such as safe behavior role model or other emotional appeals) needs a deeper analysis.

Results of this study suggest some future research directions. All previous studies, including the current, were aimed at capturing the effect of single exposure to the message. We suggest applying the condition of multiple exposure to the message in an experiment. This would ensure that drivers memorize the content of it; therefore, the effect of the intended message should be clearer. The follow-up testing could be beneficial to obtain information about long-term effects of safety messages. Researchers of future studies should look for possibilities to measure driving behavior rather than intentions as the outcome variable. The age effect as a possible moderator for the effectiveness of safety messages might be assessed in future studies, as younger drivers might be less willing to cease smartphone use while driving. In addition, future research should put more effort into unravelling consistent patterns on how different types of messages influence the effectiveness of road safety messages targeting smartphone use while driving, and what other intervening variables might modify the effect. The perception of how risky different modes of smartphone use while driving are (like texting, speaking in a hand-held phone and scrolling on social media or the internet) should be taken into account.

**Author Contributions:** Conceptualization, L.Š., A.E. and R.M.; methodology, J.S. and R.M.; software, J.S. and R.M.; validation, L.Š., A.E. and D.Š.; formal analysis, L.Š.; investigation, L.Š., R.M., J.S., D.Š. and G.T.; resources, L.Š.; data curation, A.E.; writing—original draft preparation, L.Š. and R.M.; writing—review and editing, A.E. and J.S.; visualization, D.Š.; supervision, L.Š.; project administration, L.Š.; funding acquisition, L.Š. All authors have read and agreed to the published version of the manuscript.

**Funding:** This research was funded by the grant of the scientific foundation of Vytautas Magnus University No. P-S-19-05.

**Institutional Review Board Statement:** The study was conducted according to the guidelines of the Declaration of Helsinki, and approved by the Institutional Review Board of Vytautas Magnus University No. EKL-2019.01, date of approval 12 November 2019.

**Informed Consent Statement:** Informed consent was obtained from all subjects involved in the study.

**Data Availability Statement:** The data presented in this study are available on request from the corresponding author.

**Conflicts of Interest:** The authors declare no conflict of interest.

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
