# Peer review of "Targeting Smartphone Use While Driving: Drivers’ Reactions to Different Types of Safety Messages"

_sustainability, doi:10.3390/su132313241_

Round 1
Reviewer 1 Report
This paper deals with targeting smartphone use while driving, specifically the drivers' reaction to different forms of safety messages. Although the paper is clear regarding the experiment and has provided an in-depth literature review, the presentation and the language of the paper has to be improved. 1. For example in the Abstract there are single words like Objective Methods Conclusions. These should be used as guidelines rather than explicitly mentioned in a scientific paper. The Abstract needs to be re-written to properly highlight the main contributions of the paper. 2. The introduction needs to be re-worked on such that the proper flow of different methodologies used over the years are clear. At this point, the authors have pointed to different published resources on this subject but a lack of flow makes it difficult to follow. 3. A clear contribution is missing in the paper. Dedicating a paragraph with bullet points highlighting the major contributions can help in improving the quality of the paper. 4. In the Materials and Methods section, the different important topics are divided into sub-headings. Without any pictures or visual aids, it becomes difficult to follow the paper and their subsequent results. 5. A brief comparison with other researches on this topic and how this work is useful for establishing the intended message is lacking in the results section which further confuses the reader regarding the contribution and necessity of the work. 6. It will be a good idea to add a future direction section to this work.
Author Response
Response to Reviewer 1 Comments
Comments and Suggestions for Authors
This paper deals with targeting smartphone use while driving, specifically the drivers' reaction to different forms of safety messages.
Point 1: Although the paper is clear regarding the experiment and has provided an in-depth literature review, the presentation and the language of the paper has to be improved.1. For example in the Abstract there are single words like Objective Methods Conclusions. These should be used as guidelines rather than explicitly mentioned in a scientific paper. The Abstract needs to be re-written to properly highlight the main contributions of the paper.
Response 1: The abstract was improved by adding some contribution statements and changing results section. The words “Objective, ‘Methods”, etc. were deleted. The abstract was reviewed and study relevance idea was added at the beginning. Additionally some results and implications were rewritten. The language of the manuscript has also been slightly corrected.
Point 2: The introduction needs to be re-worked on such that the proper flow of different methodologies used over the years are clear. At this point, the authors have pointed to different published resources on this subject but a lack of flow makes it difficult to follow.
Response 2: We reviewed the cited literature once again. We couldn’t find actual pattern which relates different methodologies and results, as well as historical aspects of conducting research in the area of various appeals regarding road safety messages. In revised version of manuscript, we strengthened the point that there is certain gap in research with smartphone use as targeted behavior. We added some information about methodology of previous research, and we believe it allowed better understanding of obtained results. To increase better flow of the text we added subtitles for important pieces of introduction. When showing the state of art in this area, we tried to construct the text in such an order: first, we cited just few existing studies which actually dealt with the effectiveness of messages targeting smartphone use, then, referred to other literature which used threat appeals for other driving behaviors and reported their effectiveness, then, introduced the idea that role modelling together with threat appeals in messages should increase the effectiveness. Finally, the relevance and novelty of current study was introduced.
Point 3: A clear contribution is missing in the paper. Dedicating a paragraph with bullet points highlighting the major contributions can help in improving the quality of the paper.
Response 3: The paragraph with highlights was added at the end of introduction section.
Point 4: In the Materials and Methods section, the different important topics are divided into sub-headings. Without any pictures or visual aids, it becomes difficult to follow the paper and their subsequent results.
Response 4: The figure of study design was added to the manuscript (see section 2.4).
Point 5: A brief comparison with other researches on this topic and how this work is useful for establishing the intended message is lacking in the results section which further confuses the reader regarding the contribution and necessity of the work.
Response 5: The main comparison of study results with previous literature is discussed in second and third paragraphs of discussion section. To increase the better understanding of the contribution of this study to the field, we developed further the idea about how the results of the study might be important for further safety campaigns targeting smartphone use while driving in Lithuania. Some additional highlights were added to the paragraph with implications of study results (see lines 419 – 428): as threat appeal only showed higher potential effectiveness, further use of this type of messages might be encouraged in Lithuania. This idea is now developed deeper in discussion section.
Point 6: It will be a good idea to add a future direction section to this work.
Response 6: In the previous version of manuscript future directions were provided together with limitations. It was restructured in current version. Last paragraph was dedicated to future directions.
Reviewer 2 Report
In Targeting Smartphone use while Driving: Drivers’ Reactions to different Types of Safety Message the authors test the effectiveness of two types of messages targeting smartphone use while driving among Lithuanian drivers from ongoing road safety campaign: threat appeal and threat appeal with safe behavior role modelling.
The authors should be congratulated for having addressed such a hot and public utility topic.
However, I would like to provide few suggestions as follow:
- I think in line 65 there is a typo; “Only several previous studies confirmed that”
- In Section 2.1 the authors say: “Age range was between 18 and 50 years” but in line 107 they say “had 5.9 years of driving experience in average”. Is a driving license granted to 13/14 years old people in Lithuania?
- I suggest a figure in which the framework used is emphasized, this could give the reader an overall and immediate vision of what has been done.
- In section 2.2 the authors explain the experimental protocol well, however without any new elements of novelty as the video clips have been granted by Ministry of Transport and Communications;
- Section 2.3.1 the questionnaire that they use to test the emotions of the groups is the result of previous research, rightly the paternity is recognized;
- I suggest the author to increase the content of section 2.5;
- I suggest to refocusing sections 3.1 and 3.2, the results could be addressed in a better way;
- Could these results give a different score for groups of age of subjects ?
- As stressed in section 4, this work has many limitations and furthermore it does not seem that results obtained can contribute to improving the road accident prevention system.
Honestly, I struggle to understand the novelty and the scientific robustness of this work. The authors use video clips made by the Lithuanian Ministry of Transport and the questionnaires, to test the hired subjects, taken from previous research. It seems that the only contribution is limited to a statistical analysis of the score on the questionnaires of a small group of subjects tested in a simulated environment in the lab.
Author Response
Response to Reviewer 2 Comments
Comments and Suggestions for Authors
In Targeting Smartphone use while Driving: Drivers’ Reactions to different Types of Safety Message the authors test the effectiveness of two types of messages targeting smartphone use while driving among Lithuanian drivers from ongoing road safety campaign: threat appeal and threat appeal with safe behavior role modelling.
The authors should be congratulated for having addressed such a hot and public utility topic.
However, I would like to provide few suggestions as follow:
Point 1: I think in line 65 there is a typo; “Only several previous studies confirmed that”
Response 1: Typo was corrected.
Point 2: In Section 2.1 the authors say: “Age range was between 18 and 50 years” but in line 107 they say “had 5.9 years of driving experience in average”. Is a driving license granted to 13/14 years old people in Lithuania?
Response 2: We have addressed this point by adding information about the range of driving experience to the manuscript: from 1 to 26 years, mean of driving experience – 5.9 years, SD=6.0.
Point 3: I suggest a figure in which the framework used is emphasized, this could give the reader an overall and immediate vision of what has been done.
Response 3: The figure of study design was added to the manuscript (see section 2.4).
Point 4: In section 2.2 the authors explain the experimental protocol well, however without any new elements of novelty as the video clips have been granted by Ministry of Transport and Communications;
Response 4: There is a lack of knowledge how effective are safety messages targeting smartphone use while driving for safer driving behaviors, especially those messages that are used in ongoing road safety campaigns. In Lithuania road safety messages are created using basic knowledge on social marketing and are released without any prior testing whether they have any effect and what effect might it be. Thus, authors see the novelty of this research on both focusing on illegal smart phone use while driving and testing the effectiveness of actual road safety messages. Additionally, the importance of prior behavior habits is tested. The paragraph with basic highlights and contributions of this study was added at the end of introduction section.
Point 5: Section 2.3.1 the questionnaire that they use to test the emotions of the groups is the result of previous research, rightly the paternity is recognized;
Response 5: We agree with current comment and believe that there is no need to respond further.
Point 6: I suggest the author to increase the content of section 2.5;
Response 6: Information about the statistical procedures that were used to test hypotheses was added to section 2.5
Point 7: I suggest to refocusing sections 3.1 and 3.2, the results could be addressed in a better way;
Response 7: Authors were not sure if they understood the suggestion of the reviewer, therefore they couldn’t respond to it properly. The result section was organised according to the aim of the manuscript. First, differences of evoked emotions among groups were presented, and then the effect of different message for intentions to drive and use smartphone was indicated. Finally, the effect was evaluated once more, including previous smartphone use behavior during driving as control variable. Authors couldn’t think of better way of presenting this small bit of information.
Point 8: Could these results give a different score for groups of age of subjects?
Response 8: The same statistical procedures were repeated in different age groups. Additionally, age was added as predictive variable into univariate analysis of variance. In most cases the results remained the same as reported in the manuscript. In some cases, the effect of group became insignificant. This might be related to sample size and statistical power. Sample size was appropriate for simple statistics as ANOVA and Univariate analysis of variances with two variables. Adding more variables compromise statistical power to obtain significant differences. Thus, authors decided not to add age related results to the text of manuscript. We thank the reviewer for an idea, it was added as the suggestion for future research.
Point 9: As stressed in section 4, this work has many limitations and furthermore it does not seem that results obtained can contribute to improving the road accident prevention system.
Response 9: We acknowledge all limitations of the study. Still we believe that some important implications might be drown from these few results that are important for further safety campaigns targeting smartphone use while driving. As threat appeal only showed higher potential effectiveness, further use of this type of messages might be encouraged in Lithuania. This idea is now developed deeper in discussion section (see lines 419 – 428). Additionally, the paragraph with basic highlights for novelty and relevance of this study was added at the end of introduction section as a response to Point 4.
Reviewer 3 Report
The article presents the results of an interesting experiment related to the topic of road safety. The text meets the requirements for academic papers: it is of adequate quality, appropriately cites sources, uses scientific methods and its results are well justified. I did not find any serious errors in the article. My only comment is that it would be interesting to see the video clips used in the experiment - if they are available somewhere (e.g. on Youtube), then it would be good to add this link to the article so that readers know exactly what these videos look like.
Author Response
Comments and Suggestions for Authors
Point 1: The article presents the results of an interesting experiment related to the topic of road safety. The text meets the requirements for academic papers: it is of adequate quality, appropriately cites sources, uses scientific methods and its results are well justified. I did not find any serious errors in the article. My only comment is that it would be interesting to see the video clips used in the experiment - if they are available somewhere (e.g. on Youtube), then it would be good to add this link to the article so that readers know exactly what these videos look like.
Response 1: The URL references for both safety messages were added to the manuscript, section 2.2. Both messages are in Lithuanian as were presented to the participants.
Round 2
Reviewer 1 Report
The authors have addressed all the previous concerns satisfactorily.
Reviewer 2 Report
The authors did a great job of reviewing, addressing comments and improving the content. I suggest Accept in the current form.